# Identification and Characterization of TMEM119-Positive Cells in the Postnatal and Adult Murine Cochlea

**DOI:** 10.3390/brainsci13030516

**Published:** 2023-03-20

**Authors:** Mohamed Bassiouni, Alina Smorodchenko, Heidi Olze, Agnieszka J. Szczepek

**Affiliations:** 1Department of Otorhinolaryngology, Head and Neck Surgery, Charité—Universitätsmedizin Berlin, Corporate Member of Freie Universität Berlin and Humboldt-Universität zu Berlin, Charitéplatz 1, 10117 Berlin, Germany; mohamed.bassiouni@charite.de (M.B.);; 2MSB Medical School Berlin, Rüdesheimer Str. 50, 14197 Berlin, Germany

**Keywords:** cochlea, microglia, immunofluorescence, noise exposure, macrophages

## Abstract

Transmembrane protein 119 (TMEM119) is expressed in a subset of resident macrophage cells of the brain and was proposed as a marker for native brain microglia. The presence of cells expressing TMEM119 in the cochlea has not yet been described. Thus, the present study aimed to characterize the TMEM119-expressing cells of the postnatal and adult cochlea, the latter also after noise exposure. Immunofluorescent staining of cochlear cryosections detected TMEM119 protein in the spiral limbus fibrocytes and the developing stria vascularis at postnatal Day 3. Applying the macrophage marker Iba1 revealed that TMEM119 is not a marker of cochlear macrophages or a subset of them. In the adult murine cochlea, TMEM119 expression was detected in the basal cells of the stria vascularis and the dark mesenchymal cells of the supralimbal zone. Exposure to noise trauma was not associated with a qualitative change in the types or distributions of the TMEM119-expressing cells of the adult cochlea. Western blot analysis indicated a similar TMEM119 protein expression level in the postnatal cochlea and brain tissues. The findings do not support using TMEM119 as a specific microglial or macrophage marker in the cochlea. The precise role of TMEM119 in the cochlea remains to be investigated through functional experiments. TMEM119 expression in the basal cells of the stria vascularis implies a possible role in the gap junction system of the blood–labyrinth barrier and merits further research.

## 1. Introduction

Microglia are a subset of neural glia present in the central nervous system [1,2]. In contrast to monocyte-derived recruited macrophages, resident microglia represent the native immune cells or innate macrophages of the brain [1,2]. In recent years, microglia have been shown to play an essential role in brain inflammation, aging, and homeostasis [3,4,5,6,7]. The lack of optimal markers to distinguish these two cellular entities has been a challenge in neuroscience research, although several transgenic strategies have been developed to distinguish and isolate these populations [8,9,10,11,12,13,14,15,16].

To date, many markers have been used to study the development and activation of brain macrophages and microglia [16,17,18,19,20,21], most notably ionized calcium-binding adapter molecule 1 (Iba1) [22,23]. In addition to Iba1, CX3CR1 [24,25] and other classical surface molecules, such as CD11b, CD45, F4/80, and CD68, are considered nonspecific pan-macrophage markers used to study both monocyte-derived macrophages and bona fide microglia [16,17,18,19,20,21]. In more recent years, other novel specific microglial markers have been described, such as transmembrane protein 119 (TMEM119) [9,26] and purinergic receptor P2Y, G-protein coupled 12 (P2ry12) [13,27,28]. In particular, TMEM119 appears to be the most promising microglial marker due to its abundant and highly specific expression in true microglia [9,14,26]. TMEM119 was shown to be expressed in a subset of Iba1-positive brain macrophages in mice and humans [9,26]. The findings of those studies suggested that TMEM119 can specifically distinguish resident brain microglia (TMEM119-positive) from monocyte-derived macrophages (TMEM119-negative), both of which express the classical pan-macrophage marker Iba1 [26]. In addition, brains with known Alzheimer’s disease were shown to have higher TMEM119 mRNA levels, further supporting the role of TMEM119-positive microglia in the central response to stress, inflammation, and toxicity [26]. Other less-studied novel markers include Spalt-Like Transcription Factor 1 (Sall1) [10], Hexosaminidase Subunit Beta (Hexb) [15], and Olfactomedin-Like 3 (Olfml3) [11], which are not yet widely employed in microglial research.

In recent years, increasing evidence demonstrated the existence of resident macrophages in the mammalian cochlea [29,30,31,32,33,34,35,36,37,38,39,40]. Exposure to noise trauma was shown to affect the number, distribution, and morphology of cochlear macrophages [39,41,42,43,44]. Several studies have described the expression of the classical macrophage marker Iba1 in the nonsensory cells of the mouse and human cochlea [29,30,31,32,35,38,45,46,47]. However, the expression of the microglial marker TMEM119 in the cochlea remains unknown. This study aimed to examine and characterize the TMEM119-expressing cells in the immature postnatal and adult mouse cochlea to characterize the development of cochlear microglia before and after the onset of hearing. Additionally, the effect of noise trauma on the types and distribution of TMEM119-expressing cells was investigated qualitatively using immunofluorescence to elucidate the potential role of TMEM119 in the cochlear response to inflammation and stress.

## 2. Materials and Methods

### 2.1. Animals and Tissue Harvesting

The animal experiments were approved by the Governmental Ethics Commission for Animal Welfare (LaGeSo Berlin, Germany; approval number: T 0235/18). The animals received care in compliance with Directive 2010/63/EU on the protection of animals used for scientific purposes. The C57/Bl6J mice of both sexes that initially originated from the Jackson Laboratory, were obtained from an in-house colony at the Animal Facility of Charité Universitätsmedizin Berlin. The postnatal Day 3 (p3) animals were sacrificed immediately after arrival from the Animal Facility by decapitation with a sharp pair of scissors. Adult mice were sacrificed by cervical dislocation. The entire inner ear was excised from the temporal bone and fixed with 4% paraformaldehyde (PFA) perfusion through the perilymphatic space for histological processing of the adult cochlea. Twelve animals were used for all experiments. 

### 2.2. The Noise Trauma Model

The tissues from noise trauma experiments were kindly provided by Prof. Marlies Knipper (University of Tübingen, Germany) as part of a 3R (Reduce, Reuse, Recycle) cooperation. The protocol for noise trauma was published [48]. Briefly, the mice were anesthetized and exposed to broadband noise (8–16 kHz) at a 120 dB sound pressure level for 1 h and then sacrificed seven days later. The sham-exposed control group was anesthetized without noise exposure in a sound chamber. For those two groups of mice, tissue processing proceeded according to a previously published protocol by Möhrle and coworkers [48].

### 2.3. Tissue Processing for Cryosections

The cochleae were incubated in 4% PFA for 2 h at 4 °C. Subsequently, the PFA solution was washed out with 1x PBS, followed by incubation in 1x PBS solution three times for 5 min each time. The cochleae of adult mice were decalcified in 10% ethylenediaminetetraacetic acid (EDTA) (Sigma‒Aldrich, Darmstadt, Germany) for 24 h at 4 °C. Subsequently, the perilymphatic space was perfused with 1x PBS to wash out the EDTA solution. The half-heads of p3 mice did not require decalcification. The p3 half-heads or the isolated adult inner ear specimens were incubated at 4 °C overnight in 30% sucrose in 1x PBS for cryoprotection. The following day, the specimens were cryo-embedded in PolyFreeze medium (Sigma‒Aldrich) and sectioned at 10 µm thickness using a Leica CM3050 cryostat (Leica Biosystems, Wetzlar, Germany). The tissue sections were stored at −20 °C until immunolabeling was performed.

### 2.4. Immunolabeling and Confocal Microscopy

TMEM119 immunofluorescence analysis was performed on cochlear cryosections permeabilized with 0.1% Triton X-100 (Sigma‒Aldrich) in 1x PBS and subsequently incubated with 5% normal goat serum (Jackson Dianova, Hamburg, Germany) or normal donkey serum (Aviva Systems Biology, San Diego, CA, USA) in PBS at room temperature for 30 min. The primary antibodies used included rabbit monoclonal anti-TMEM119 antibody (1:200, ab209064; Abcam, Cambridge, UK), rabbit anti-Iba1 (1:500, Wako Chemicals, Neuss, Germany), goat anti-Iba1 (1:200, ab5076, Abcam), and mouse anti-claudin-11 (1:100, sc-271232, Santa Cruz Biotechnology, Heidelberg, Germany). The cryosections were incubated with the primary antibodies overnight at 4 °C. During the immunofluorescence labeling with the mouse anti-claudin-11 antibody, to block the mouse-on-mouse background, an extra blocking step with anti-mouse IgG blocking reagent (Vector Laboratories, Burlingame, CA, USA) was performed, as described previously [49]. The secondary antibodies that were used were goat anti-mouse 568 (Thermo Fisher Scientific, Darmstadt, Germany, A-11004), goat anti-rabbit 488 (Thermo Fisher Scientific, A-11034), donkey anti-goat DyLight 594 (Abcam, ab96937), and donkey anti-rabbit DyLight 488 (Novus Biologicals, NBP1-75292). The secondary antibodies were incubated in the dark for 1 h at room temperature. Finally, the slides were counterstained with DRAQ5 fluorescent dye (Thermo Fisher Scientific) to visualize cellular DNA, coverslipped using ProLong™ Gold Antifade Mountant with DAPI (Thermo Fisher Scientific), and then stored in the dark at 4 °C until microscopy. The confocal imaging of sections stained with TMEM119, DRAQ5, and Iba-1 was performed using a Leica TCS SL confocal microscope (Leica Biosystems) with immersion oil objectives x40 and x63. The Alexa-488 was excited using argon laser (488 nm), and the Alexa-594 and DRAQ5 with helium–neon laser (543 nm and 633 nm, respectively).

The confocal imaging of specimens stained with TMEM119, Iba-1, Claudin-11, and DAPI was performed using a Leica confocal microscope STELLARIS 5 (Leica Biosystems, Wetzlar, Germany) with a water objective x40 and oil objective x63. The Alexa-488 and Alexa-594 were excited with integrated White Light Laser (WLL) and DAPI (405 nm) with laser 405 DMOD.

### 2.5. Protein Isolation and Western Blot Analysis

The membranous cochlear tissue was microdissected and harvested from the postnatal C57/Bl6J mouse cochlea, as described previously [50]. After separating the lateral wall containing stria vascularis from the cochlear duct containing the organ of Corti, the isolated tissues were stored separately. Postnatal mouse brain and adult mouse spleen tissues were harvested and used as positive and negative controls, respectively. The Western blot protocol was previously described [51]. Briefly, the tissue lysates were prepared by placing and vortexing the explants in 80 µL RIPA buffer (Cell Signaling Technology, Danvers, MA, USA) and then centrifuging them at 14,000 RPM at +4 °C for 10 min. For every aliquot, a total protein amount of 14–16 µg was mixed with the Roti-Load sample loading solution (Carl Roth GmbH, Karlsruhe, Germany) and heated at 90 °C for 5 min in a thermomixer (Eppendorf, Hamburg, Germany). The samples were loaded in Novex WedgeWell 4–20% Tris-Glycine Mini Gels (12 well; Thermo Fischer Scientific) followed by electrophoresis using an XCell SureLock^®^ Mini-Cell gel electrophoresis system (Thermo Fischer Scientific) at 180 V for 65 min. PageRuler Plus Prestained Protein Ladder (Thermo Fischer Scientific) was used as a protein marker. After electrophoresis, the proteins were blotted onto a 0.45 µm Immobilon-P Transfer Membrane (Thermo Fischer Scientific) at 300 mA for 44 min (Biometra GmbH, Göttingen, Germany). The membranes were blocked with 5% skimmed milk powder solution prepared in PBS and containing 0.1% Tween 20 (Sigma‒Aldrich) for 1 h at room temperature, followed by incubation for 2 h with the primary rabbit antibody against TMEM119 (1:500, cat # PA5-119902, Thermo Fischer Scientific) or ß-Actin (1:5000, cat # A1978, clone AC-15, Sigma‒Aldrich). Following several washes and incubation with the secondary antibodies, the signal was detected by incubation with the SuperSignal West Femto Maximum Sensitivity Substrate (Thermo Fisher Scientific) and then direct measurement of chemiluminescence using a C-Digit scanner (LI-COR Biotechnology-GmbH, Bad Homburg vor der Höhe, Germany). GelScan Pro V.6.0 software was used for the quantification.

### 2.6. RNA Isolation and Semiquantitative Real-Time Reverse-Transcription–Polymerase Chain Reaction (RT-PCR)

To isolate the total RNA, RNeasy Mini Kit was used strictly according to the manufacturer’s instructions (cat. # 74106, Qiagen, Hilden, Germany). The RNA was isolated from the brain, spleen, and entire membraneous cochlea of p3 mice, and its concentration was measured using NanoDrop One (cat. # 701-058112, Thermo Fisher Scientific, Hennigsdorf, Germany). Three samples from each tissue were processed (biological replicates). The purity of obtained RNA was assessed using the ratio A_260_/A_280_, which, in all cases, was between 2.0 and 2.1. 

The one-step real-time RT-PCR was performed using QuantiNova™ SYBR Green RT-PCR Kit (cat. # 208154, Qiagen) with primers targeting TMEM119 (Mm_Tmem119_1_SG QuantiTect Primer Assay; GeneGlobe ID—QT00256025, Qiagen) and beta-actin as a housekeeping gene (Mm_Actb_1_SG QuantiTect Primer Assay; GeneGlobe ID—QT00095242, Qiagen). For each reaction well on the PCR 96-well plate (cat. # 04729692001, Roche Deutschland Holding GmbH, Mannheim, Germany), 10 µL of 2x SYBR Green RT-PCR Master Mix, 0.2 µL QN SYBR Green RT-Mix, 2 µL of 10x QuantiTect Primer Assay, 100 ng of total RNA, and PCR-quality H_2_O up to 20 µL of the final volume was added. After that, sealing foil was secured on the plate, which was spun down at 1000 rpm for 1 min. LightCycler® 96 (cat. # 05815916001, Roche) was set to perform the reverse transcription reaction and PCR in the following way: 50 °C for 10 min (reverse transcription step) followed by 2 min at 95 °C (activation of DNA polymerase) and 45 cycles of annealing and extension at 60 °C for 30 s and denaturation at 95 °C for 15 s. The run ended with a melting step (10 s at 95 °C, 60 s at 65 °C, and 1 s at 97 °C) followed by cooling up to 37 °C. Data were acquired using Instrument Software V1.2 (Roche) setup for SybrGreen. 

### 2.7. Statistical Analysis

The statistical analysis was performed using GraphPad software (https://www.graphpad.com/, last accessed on 8 March 2023).

## 3. Results

### 3.1. Localization of TMEM119 Protein in the Immature Postnatal Murine Cochlea

In the immature cochlea of the p3 mice, TMEM119 was detected in the basal cell region of the stria vascularis and the fibrocytes of the spiral limbus (Figure 1) (*n* = 4 mice). 

TMEM119 was not detected in the vicinity of the organ of Corti. TMEM119 localization contrasted with that of Iba1 on macrophages scattered across various cochlear regions (Appendix A). Double labeling with an anti-Iba1 antibody revealed markedly different localization patterns, with only minor overlap among cells of the spiral limbus fibrocytes and the stria vascularis (Figure 2). The results indicated that TMEM119-positive cells do not express macrophage markers in the postnatal murine cochlea nor represent a subset of Iba1-positive macrophages.

### 3.2. Localization of TMEM119 Protein in the Adult Murine Cochlea under Basal and Noise-Exposed Conditions

To determine the effect of noise trauma on TMEM119 expression, cochlear sections from noise-exposed mice were immunolabeled for TMEM119 and compared with the cochleae of sham-exposed mice (*n* = 5 mice per group). Unexposed (freshly harvested) and sham-exposed adult C57/Bl6J mice cochleae had an identical staining pattern (data not shown). The TMEM119 protein was detected in the stria vascularis and was restricted only to the region of the basal cells (Figure 3). Additionally, a specific signal was detected in the dark mesenchymal cell region of the supralimbal zone (Figure 3). Again, no TMEM119 was observed in the organ of Corti or spiral ganglion regions of the adult cochleae (Figure 3). Co-labeling with the basal cell marker claudin-11 confirmed that, in the adult stria vascularis, TMEM119 is expressed explicitly in the basal cells (Figure 4).

Noise trauma appeared to have no visible effect on the TMEM119 localization pattern in the stria vascularis, which remained localized to the basal cells, showing an indistinguishable pattern in the noise-exposed and sham-exposed groups (Figure 5A,B) (*n* = 5 each). Similarly, noise exposure appeared to cause no gross change in the expression of TMEM119 in the supralimbal zone (Figure 6A,B, *n* = 5 each). 

### 3.3. Quantitative Comparison of the TMEM119 Protein Levels in the Postnatal Cochleae and the Brain

To compare TMEM119 protein expression in the cochlea and the brain at the quantitative level, cochlear duct and stria vascularis from postnatal C57/Bl6J mice were used. Western blot analysis revealed similar TMEM119 levels in the brain, cochlea, and stria vascularis (Figure 7A). A very faint signal was seen in the spleen lysates (negative control), confirming the assay’s specificity. These results indicate similar TMEM119 protein levels in the cochlea comparable to those in the brain. The densitometry revealed no differences among the brain, cochlear, or strial TMEM119 signal intensity (Figure 7B). 

### 3.4. TMEM119 Gene Expression in the Cochlear Tissues

To determine whether the TMEM119 gene is expressed in the cochlear tissues, we performed one-step RT-PCR using total RNA isolated from the entire membraneous cochlea, including spiral limbus, organ of Corti, spiral ganglion, and lateral wall. Each sample represented two cochleae isolated from one mouse. Three biological samples, each obtained from a separate animal, were assayed in duplicates for the expression of TMEM119 (target sequence accession number NM_146162) and a housekeeping gene β-actin (target sequence accession number NM_007393). The obtained Ct and calculated ΔCt values (Table 1) indicated TMEM119 mRNA expression in the cochlear tissues. 

An independent-samples *t*-test was conducted to compare ΔCt of TMEM119 in the cochlea and brain and the cochlea and spleen. There was a significant difference in the scores for cochlea (M = 11.80, SD = 0.24) and spleen (M = 15.53, SD = 0.98); t(4) = 6.3947, *p* = 0.0031 as well as cochlea and brain (M = 16.83, SD = 0.11); t(4) = 33.49, *p* < 0.0001. When the average ΔCt obtained for the brain was used as a control, ΔΔCt values indicated that in the cochlea, the number of mRNA molecules encoding TMEM119 was, on average, 33 times greater than in the brain (SD = 3.8). 

## 4. Discussion

The existence of resident immune cells in the cochlea has been shown in numerous studies [29,30,31,32,33]. Cochlear resident macrophage cells expressing Iba1 were demonstrated in the murine [30,32,38,47] and human cochlea [29,31,33,52]. In a recent study by Okayasu et al. [33], the number of Iba1-positive macrophages in the human cochlea increased after cochlear implantation, suggesting an inflammatory response to the implantation trauma [33]. In addition, noise trauma increased the number of cochlear macrophages [39,41,42,43,44]. Furthermore, macrophage regulation and activation have been implicated in the cochlear response to ototoxicity [53] and aging [52].

In the auditory system, the nomenclature of macrophages vs. microglia has shown frequent discrepancies that may lead to confusion. Macrophages expressing classical markers, such as Iba1, CD45, and CX3R1, were frequently referred to as “microglia-like cells” in previous studies of the peripheral [35,47,54,55,56,57] and central auditory systems [58,59]. However, some authors have proposed that microglia-like cells of the auditory system are unlikely to be true microglia [56]. Nevertheless, the term “microglia” is still occasionally used in the literature to describe cells of the central auditory system expressing conventional pan-macrophage markers, such as Iba1, CD45, CD68, and CD11b [60,61,62,63]. Thus, the distinction between the terms “macrophages” and “microglia” appears less established in the auditory research literature than in the CNS literature, in which those two populations are clearly recognized as separate entities. Indeed, this discrepancy motivated the development of TMEM119 transgenic mice: the goal was to distinguish between monocyte-derived macrophages and true brain microglia [9,14,26]. The findings of those previous studies encouraged us to perform a qualitative immunohistochemical study of the TMEM119 protein in the developing and adult mouse cochlea.

In the brain, innate tissue-resident microglia are marked by TMEM119 and represent a subset of Iba1-positive macrophages [9,26]. Thus, Satoh et al. [26] hypothesized that TMEM119 represents a valuable marker for tissue-derived microglia, distinguishing them from blood-derived recruited macrophages. 

In the present study, the presence of TMEM119-positive cells was shown for the first time in the immature postnatal and adult mouse cochlea using a knockout-validated commercial antibody. We did not detect significant numbers of TMEM119 or Iba1 double-positive cells in the cochlea, and TMEM119 was largely absent on the scattered Iba1-positive macrophages. Instead, TMEM119 was consistently detected in particular cell types in the stria vascularis and spiral limbus. These findings essentially decouple TMEM119 from Iba1-expressing macrophages in the cochlea, contrasting with their established association in brain microglia. In the retina, Su et al. [64] showed that TMEM119 expression did not specifically mark the microglial cells of the retina. The authors of that study concluded that although TMEM119 is an excellent specific marker of microglia in the brain, it does not appear to be a useful microglial marker in the retina [64]. In the present study, our findings do not support using TMEM119 as a microglial marker in the cochlea.

The levels of TMEM119 protein detected in the cochlea of p3 mice were similar to those in the brain. However, the relative levels of mRNA encoding TMEM119 were significantly higher in the cochlea than in the brain. The differences between tissues concerning protein and mRNA levels are well known in the inner ear tissues and explained by tissue- or cell-specific translation rate and protein degradation level [65]. The international database for the laboratory mouse MGI (https://www.informatics.jax.org/marker/MGI:2385228, accessed on 8 March 2023) reports the expression of TMEM119 in the cochlea of P4 and P4-to-adult mice, as per whole genome sequencing. In general, the TMEM119 gene expression is reported as ubiquitous in the murine tissues as well as in the human tissues and organs (https://www.genecards.org/cgi-bin/carddisp.pl?gene=TMEM119#expression, accessed on 9 March 2023). The developmental changes in murine TMEM119 gene and protein expression in different organs and tissues have yet to be studied, and our results justify further research in that area. 

In the present study, the introduction of noise damage did not grossly affect the qualitative pattern or distribution of TMEM119-expressing cells in the cochlea. Therefore, it is challenging to speculate about the function of TMEM119 in the cochlea. The cochlear mesenchyme and lateral wall have been previously shown to be common sites of noise-induced and age-associated inflammation and, thus, potential therapeutic targets [66]. In particular, the basal cells of the stria vascularis are the site of the tight junctions that constitute the blood–labyrinth barrier [67,68]. The effect of cisplatin ototoxicity on the stria vascularis has been attributed to the disruption of the gap junctions and the blood–perilymph barrier, resulting in fibrosis, inflammation, and macrophage activation [69]. Breglio et al. [70] described the persistent accumulation of cisplatin in the stria vascularis long after the ototoxic insult itself, further emphasizing the role of the stria vascularis as a potential therapeutic target in ototoxicity. In summary, the findings may indicate a potential role of TMEM119 in the cochlear response to toxicity and inflammation in the stria vascularis. Further studies are needed to better characterize the function of TMEM119 in the cochlea with loss-of-function models and fate-mapping transgenic approaches to elucidate its potential role in cochlear homeostasis and pathology.

## Figures and Tables

**Figure 1 brainsci-13-00516-f001:**
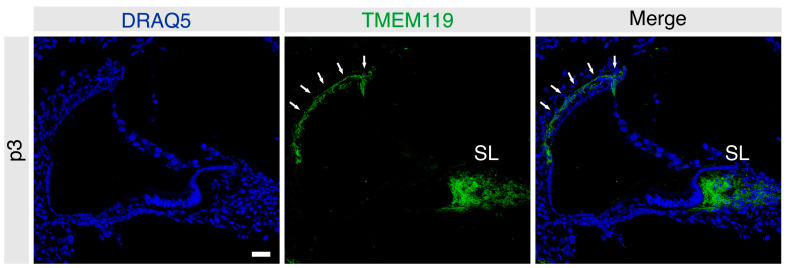
Frozen sections of the immature mouse cochlea at p3 labeled for TMEM119 (green). Nuclei were counterstained with DRAQ5 (blue). TMEM119 protein expression was detected in the intermediate and basal cells of the stria vascularis (solid white arrows) and the fibrocytes of the spiral limbus (SL) at p3. Scale bar: 20 µm.

**Figure 2 brainsci-13-00516-f002:**
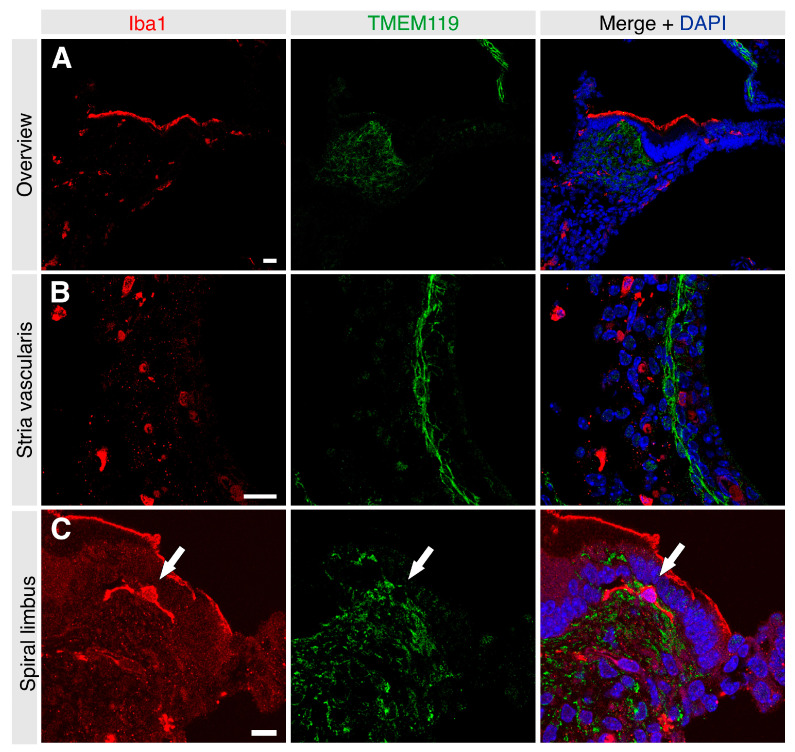
Frozen sections of the immature mouse cochlea at p3 were labeled for both Iba1 (red) and TMEM119 (green). Nuclei were counterstained with DAPI (blue). (**A**) TMEM119 was detected in the spiral limbus and cochlear lateral wall. Iba1-positive cells were detected in the stria vascularis and among spiral ligament fibrocytes, showing no obvious pattern of overlap with TMEM119-positive cells. Scale bar: 20 µm. (**B**) TMEM119 signal (green) was detected in the intermediate and basal cell layers of the stria vascularis, showing no overlap with Iba1-positive cells of the lateral cochlear wall. Scale bar: 20 µm. (**C**) TMEM119 protein expression was detected among most fibrocytes of the spiral limbus at p3. Some scattered double-positive cells were detected in the spiral limbus (arrow). Scale bar: 10 µm.

**Figure 3 brainsci-13-00516-f003:**
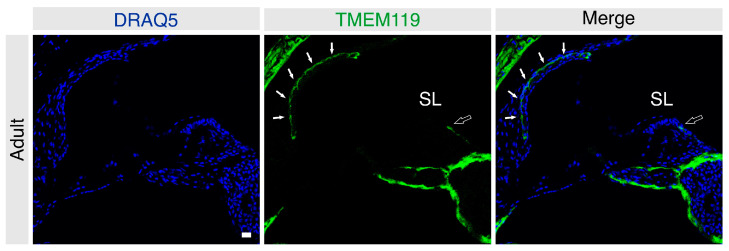
Frozen sections of the adult mouse cochlea labeled for TMEM119 (green). Nuclei were counterstained with DRAQ5-labeling (blue). TMEM119 protein expression was detected in the basal cells of the stria vascularis (solid white arrows) in the adult mouse cochlea. A specific signal was also observed in the mesenchymal dark cells of the supralimbal region (hollow arrows). The cochlear bone shows a strong fluorescence signal. Scale bar: 20 µm.

**Figure 4 brainsci-13-00516-f004:**
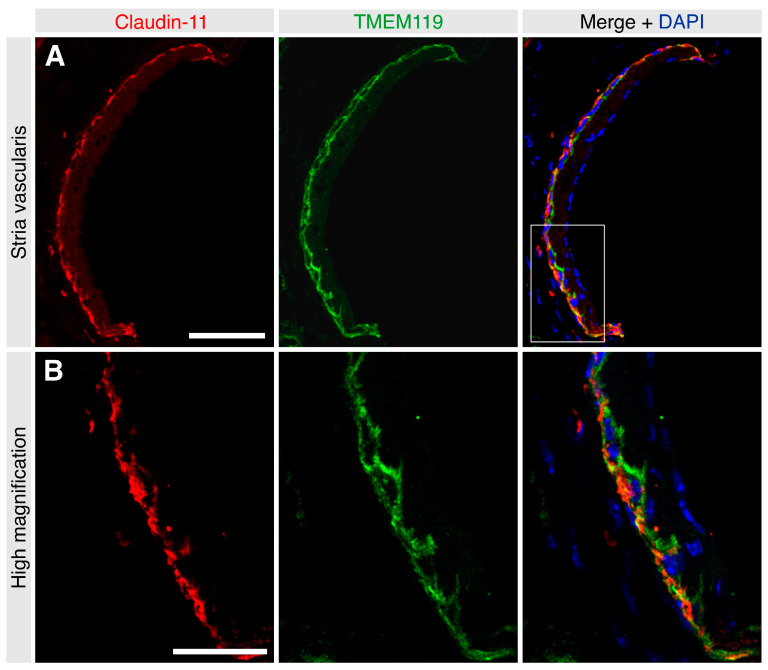
Frozen sections of the adult mouse stria vascularis labeled for TMEM119 (green) and claudin-11 (red). Nuclei were counterstained with DAPI (blue). (**A**) TMEM119 protein expression was detected in the basal cells of the stria vascularis colocalizing with the gap junction protein claudin-11. Scale bar: 50 µm. (**B**): The inset represents the high magnification image, which shows the colocalization between the TMEM119 cytoplasmic signal (green) and the claudin-11 signal (red) that appears concentrated in the basolateral aspect of the strial basal cells. Scale bar: 20 µm.

**Figure 5 brainsci-13-00516-f005:**
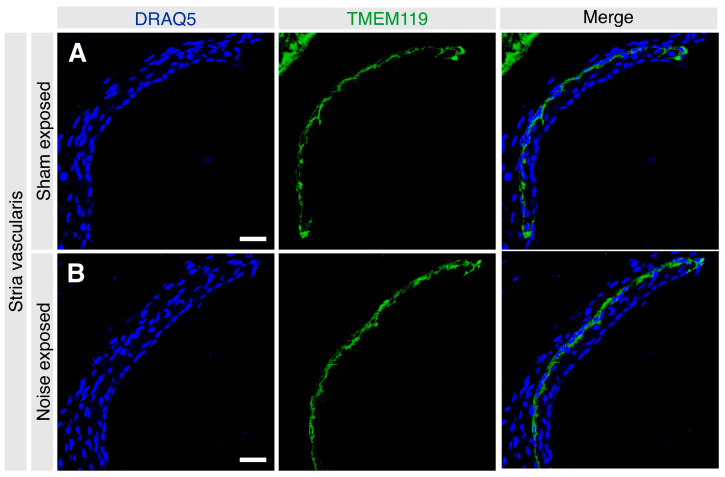
High magnification confocal microscopic images of the stria vascularis in sham-exposed (**A**) and noise-exposed (**B**) adult cochleae labeled for TMEM119 (green) and counterstained with DRAQ5 (blue). TMEM119 protein was detected in both groups’ basal cells of the stria vascularis seemingly without any detectable difference. Scale bar: 20 µm.

**Figure 6 brainsci-13-00516-f006:**
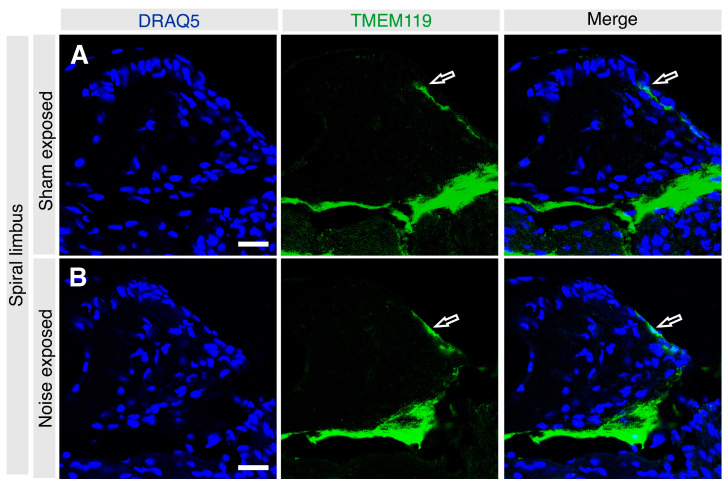
High magnification confocal microscopic images of the spiral limbus in sham-exposed (**A**) and noise-exposed (**B**) adult cochleae labeled for TMEM119 (green) and counterstained with DRAQ5 (blue). TMEM119 protein was detected in both groups’ mesenchymal cells of the supralimbic zone (hollow white arrows). Scale bar: 20 µm.

**Figure 7 brainsci-13-00516-f007:**
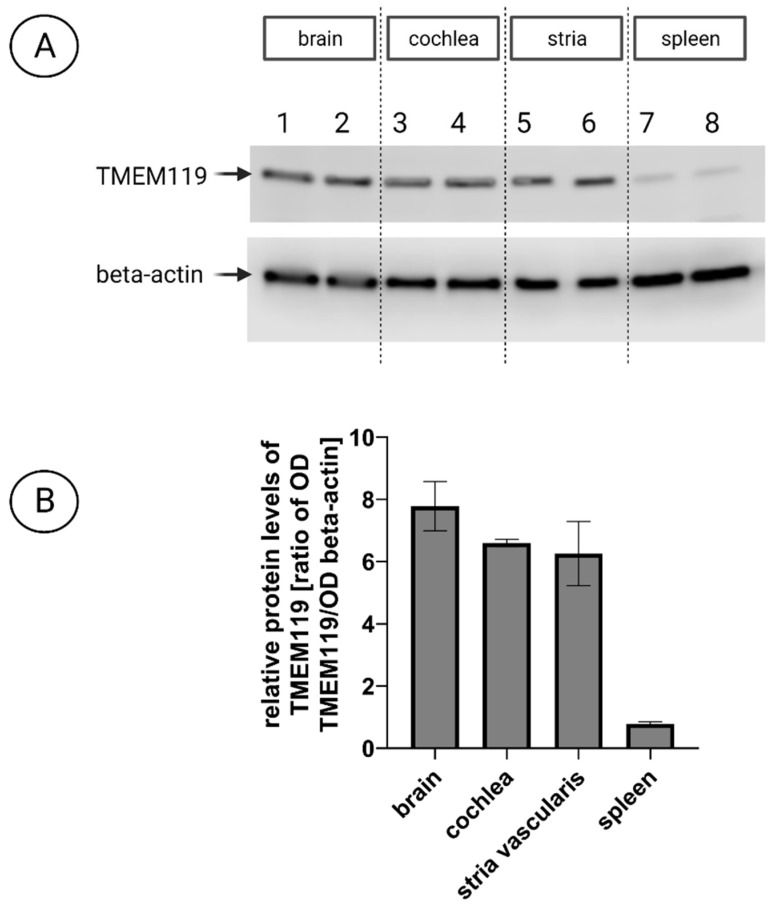
The levels of TMEM119 protein in the cochlea and stria vascularis of postnatal C57/Bl6J mice. (**A**) Representative Western blot (WB) images showing TMEM119 (MW 40 kDa) protein in the cochlea (lines 3 and 4) and stria vascularis explants (lines 5 and 6) compared with the brain as a positive control tissue (lines 1 and 2) and spleen as negative control (lines 7 and 8). Β-actin (42 kDa) was used as a loading control for all samples; 16 µg of protein per lane was loaded. The quantitative densitometry analysis of TMEM119 protein relative to β-actin revealed similar TMEM119 protein levels in the brain, cochlea, and stria vascularis (between 6- and 8-fold relative to β-actin). The spleen had much lower TMEM119 protein levels (approximately 1.03-fold relative to β-actin). (**B**) Relative TMEM119 levels were calculated on the basis of total protein staining intensity per line (reversible Ponceau S staining) and the optical density of bands representing β-actin and TMEM119. An independent-samples *t*-test determined no statistical differences between the levels of TMEM119 protein in the cochlea and brain or between the cochlea and stria. There were significant differences between TMEM119 levels in the spleen and brain (*p* = 0.0064), spleen and stria vascularis (*p* = 0.0174), and spleen and cochlea (*p* = 0.0003).

**Table 1 brainsci-13-00516-t001:** Crossing threshold (Ct) values acquired during the real-time RT-PCR for the housekeeping gene β-actin andTMEM119.

Sample	Ct Beta-Actin (Duplicate 1)	Ct Beta-Actin (Duplicate 2)	Average Ct (Beta-Actin)	Ct TMEM119 (Duplicate 1)	Ct TMEM119 (Duplicate 2)	Average Ct (TMEM119)	ΔCt (Average Ct TMEM119—Average Ct Beta-Actin)
cochlea 1	20.69	20.33	20.51	32.95	32.20	32.58	12.07
cochlea 2	20.28	20.28	20.28	32.17	31.77	31.97	11.69
cochlea 3	20.13	20.06	20.10	32.00	31.47	31.74	11.64
brain 1	20.36	18.95	19.66	36.62	36.19	36.41	16.75
brain 2	19.44	19.23	19.34	36.97	35.61	36.29	16.96
brain 3	19.44	19.20	19.32	36.24	35.97	36.11	16.79
spleen 1	19.46	18.98	19.22	34.46	32.88	33.67	14.45
spleen 2	19.62	19.50	19.56	35.91	34.77	35.34	15.78
spleen 3	19.51	19.36	19.44	36.37	35.23	35.80	16.37

## Data Availability

Supporting data may be obtained from the authors upon request.

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
