# Peer review of "Identification and Characterization of TMEM119-Positive Cells in the Postnatal and Adult Murine Cochlea"

_brainsci, 2023, doi:10.3390/brainsci13030516_

Round 1

Reviewer 1 Report

The work conducted by Bassiouni et al. evaluated the expression of TMEM119 in the postnatal and adult cochlea, the latter also after noise exposure and compared the labeling of this protein with IBA1 in various tissues in this process. Although the study has scientific relevance, there are some details that could be improved for its acceptance, which are described below:

- In the introduction, the authors should present the relationship between noise trauma and the expression/activation of microglia in the cochlea. This sentence is missing for the reader's understanding;

- The C57BL6J animals are sourced from The Jackson Laboratory. If not, as described in the text, remove the "J" from the lineage name;

- In the IHC methodology, the number of micrometers the tissues were cut was missing and the scale of the image was missing;

- Dilution of rabbit anti-IBA1 (Wako) is not described in the text;

- Why in the IHC experiments did the authors not use the same nuclei markers?

- Even though there was no difference in most of the evaluated tissues, it is important that the authors present the statistical analysis performed in bar graphs that compared the expression of proteins for TMEM119 with B-actin. It appears that TMEM119 levels are reduced in the spleen.

Author Response

We thank the Reviewers for their consideration, time, and valuable remarks that helped improve the manuscript. Below are the point-by-point responses to the concerns raised. All the changes in the text are marked.

“The work conducted by Bassiouni et al. evaluated the expression of TMEM119 in the postnatal and adult cochlea, the latter also after noise exposure and compared the labeling of this protein with IBA1 in various tissues in this process. Although the study has scientific relevance, there are some details that could be improved for its acceptance, which are described below:

- In the introduction, the authors should present the relationship between noise trauma and the expression/activation of microglia in the cochlea. This sentence is missing for the reader's understanding;”

We thank the Reviewer for this helpful suggestion. Noise trauma has been shown to increase the number and alter the distribution and morphology of cochlear macrophages. We now add this sentence in the introduction for the reader’s understanding and cite the relevant literature:

“In recent years, increasing evidence demonstrated the existence of resident macrophages in the mammalian cochlea [29-40]. Exposure to noise trauma was shown to affect the number, distribution, and morphology of cochlear macrophages [39,49-52]. Several studies have described the expression of the classical macrophage marker Iba1 in the nonsensory cells of the mouse and human cochlea [29-32,35,38,41-43]. However, the expression of the microglial marker TMEM119 in the cochlea remains unknown.”

“The C57BL6J animals are sourced from The Jackson Laboratory. If not, as described in the text, remove the "J" from the lineage name;”

The mice used in this study were of the C57/Bl6J lineage. Our animal facility purchased them from the Jackson Laboratory. The colony is maintained and bred by the facility. We were using the sixth generation.

“In the IHC methodology, the number of micrometers the tissues were cut was missing and the scale of the image was missing;”

Thank you for pointing this out. We now mention the slice thickness in the IHC methodology (10 µm).

“Dilution of rabbit anti-IBA1 (Wako) is not described in the text;”

The dilution of the rabbit anti-Iba1 antibody from Wako is now mentioned in the Methods section of the manuscript (1:500).

“Why in the IHC experiments did the authors not use the same nuclei markers?”

We appreciate this remark. The majority of presented experiments were performed between 2020 and 2022. The reasons for using two different nuclear stains were canceled orders or extremely long delivery times for several reagents during the pandemic. We used what we had in the laboratory or shared with the neighboring labs. However, we believe that using two different reagents to visualize the nuclei does not influence the results since DAPI and DRAQ5 are established nuclear markers.

“Even though there was no difference in most of the evaluated tissues, it is important that the authors present the statistical analysis performed in bar graphs that compared the expression of proteins for TMEM119 with B-actin. It appears that TMEM119 levels are reduced in the spleen.”

We thank the Reviewer for this valid point. We now include the quantitative densitometry data demonstrating the protein expression of TMEM119 in the different tissue samples relative to the basal ß-actin expression. The main message of this Figure was to show the presence of TMEM119 protein using a method other than IF.

Reviewer 2 Report

This manuscript evaluated the expression of transmembrane protein 119 (TMEM119) in cochear tissues, and showed the patterns in the noise exposure model with no significant changes. The results of this manuscript will provide useful information for basic otological researchers, whereas there are some concerns before the publish of this manuscript.

Please provide graphs and statistical evaluations of the image (area, intensity or whatever) to show the difference. Only numbers of mice described in the texts are insufficient to support the discussion.

Please also provide qPCR information on the expression of TMEM119. The western blot results only may be insufficient to support the expression pattern in tissues.

Please provide the typical microdissection images to extract the tissues in cochlea and stria vascularis.

Author Response

We thank the Reviewers for their consideration, time, and valuable remarks that helped improve the manuscript. Below are the point-by-point responses to the concerns raised. All the changes in the text are marked.

“This manuscript evaluated the expression of transmembrane protein 119 (TMEM119) in cochear tissues, and showed the patterns in the noise exposure model with no significant changes. The results of this manuscript will provide useful information for basic otological researchers, whereas there are some concerns before the publish of this manuscript.

Please provide graphs and statistical evaluations of the image (area, intensity or whatever) to show the difference. Only numbers of mice described in the texts are insufficient to support the discussion.”

We appreciate your raising this concern. We present qualitative data concerning the “spatial” expression pattern of TMEM119-positive cells and quantitative data regarding the protein level and gene expression. We believe that immunofluorescence labeling of cochlear sections is a qualitative method to show the cell types expressing TMEM119 in the cochlea. We do not claim that noise damage leads to an upregulation or downregulation of TMEM119 but that the cell types and distribution of TMEM119-expressing cells remain unchanged. We mention the number of animals used to support this qualitative statement.

To clearly define the aim of the manuscript, we have changed the title to “Identification and characterization of TMEM119-Positive Cells in the Postnatal and Adult Murine Cochlea”.

We have also replaced the term “TMEM119 expression” with “TMEM119-expressing cells” in the manuscript to prevent the misleading notion (our fault!) of making quantitative claims about TMEM119 expression on a single cell level, as per immunofluorescence. We thank the Reviewer for this comment, which helped improve the manuscript. We hope the Reviewer agrees that our qualitative immunofluorescence analysis is sufficient to support our conclusion about the cell types expressing TMEM119, which remain the same after noise damage.

Please also provide qPCR information on the expression of TMEM119. The western blot results only may be insufficient to support the expression pattern in tissues.

We have followed this suggestion and performed a real-time semi-quantitative one-step RT-PCR to assess the relative expression of TMEM119 in the cochlea, brain and spleen. We have added RT-PCR-dedicated subsections to address the new data in the Methods, Results, and Discussion.

Please provide the typical microdissection images to extract the tissues in cochlea and stria vascularis.

The suggestion is appreciated; however, we recently published our detailed microdissection methodology in a method-dedicated article (Bassiouni, M.; Stölzel, K.; Smorodchenko, A.; Olze, H.; Szczepek, A.J. Tackling the mouse-on-mouse problem in cochlear immunofluorescence: A simple double-blocking protocol for immunofluorescent labeling of murine cochlear sections with primary mouse antibodies. Curr. Protoc. Mouse Biol. 2020, 10, e84; https://doi.org/10.1002/cpmo.84) to which we refer in our present manuscript. That work contains figures and photographs of the dissection and isolation of the cochlea. No new microdissection method was developed for the present work.

Round 2

Reviewer 2 Report

The most concerns raised by the reviewer seemed to have been completed.